# Climate-sensitive zoonotic diseases transmissible by companion animals: A scoping review protocol

Samantha Hobson[1], John Mallare[1], Heather Grieve[1], J. Scott Weese[2], Lauren E. Grant[1]*

**1** Department of Population Medicine, Ontario Veterinary College, University of Guelph, Guelph, Ontario, Canada, **2** Department of Pathobiology, Ontario Veterinary College, University of Guelph, Guelph, Ontario, Canada

* laugrant@uoguelph.ca

## Abstract

### Objective

This joint protocol describes two scoping reviews that will identify and describe evidence for climate sensitivity of companion animal zoonotic diseases in cat, dog, and human populations worldwide.

### Introduction

Climate change is a driver for emerging and re-emerging zoonotic diseases of global health concern. Companion animals can transmit over 70 zoonotic pathogens, some of which are sensitive to changes in meteorological factors. There is disparate evidence in our understanding of climate-sensitive companion animal zoonotic diseases.

### Inclusion criteria

Primary research articles that describe 1) an association or effect between meteorological factors and the risk of zoonotic disease, 2) the presence of spatiotemporal variations in disease incidence or prevalence, or 3) the projected impacts of climate emission scenarios on disease trends will be included.

### Methods

A comprehensive search strategy was developed using index terms and keywords for populations of interest, companion animal zoonotic diseases, and meteorological factors. Articles will be searched on MEDLINE (via Ovid), AGRICOLA (via ProQuest), and Web of Science. Additional articles will be identified using citation tracking. Independent reviewers will systematically apply a two-step study screening process based on defined eligibility criteria. Key study characteristics and findings

**Data availability statement:** No datasets were generated or analysed during the current study. All relevant data from this study will be made available upon study completion.

**Funding:** This work is supported by a grant to LEG from the Public Health Agency of Canada Infectious Disease and Climate Change Fund Program (2324-HQ-000026) (https://www.canada.ca/en/public-health.html). The funder played no role in study design, decision to publish, or preparation of the manuscript.

**Competing interests:** The authors have declared that no competing interests exist.

will be collated and presented as a descriptive summary using graphical and tabular formats.

## Review registration

Two separate protocols have been registered in Open Science Framework. The first review consolidates evidence in cat and dog populations (https://osf.io/ydgc2), while the second review is focused on human populations (https://osf.io/3cvx2).

## Introduction

Zoonotic diseases are characterized by their ability to be transmitted between animals and people, caused by pathogens that can infect both animals and humans [1]. Zoonotic pathogens are found across all major taxonomic groups, including fungi, helminths, protists, viruses, and bacteria [2]. Similar to other infectious pathogens, zoonotic pathogens can be climate-sensitive, and some may be more sensitive to climate than animal- or human-only diseases [2]. Climate-sensitive pathogens and diseases are characterized by their susceptibility to fluctuations in temperature, precipitation, and extreme weather events. Changes in temperature and precipitation can alter the distribution of disease vectors and reservoirs, while all three factors can further alter migration and mobility patterns [3–6]. Through these mechanisms, climate change has enabled certain zoonotic pathogens to emerge in new geographic areas or re-emerge in previously established areas, increasing the risk of disease [7–9]. Climate change is expected to increase the risk of zoonotic diseases via changes in temperature, precipitation patterns, extreme weather events, and other meteorological factors [2,7,8,10].

The impacts of climate change on various zoonotic diseases can be seen globally, across all taxonomic groups of pathogens, and multiple species including wildlife, domestic animals, and humans [2,10,11]. Examples of climate-sensitive zoonotic diseases include Lyme disease [7], leishmaniasis [12,13], rabies [14], leptospirosis [15], and anthrax [16]. Companion animals, restricted to cats (i.e., *Felis catus)* and dogs (i.e., *Canis familiaris)* for the purpose of this protocol, may act as sentinels for some zoonotic diseases. Their shared living environment with human owners, interactions with wildlife, and free-roaming nature in many areas worldwide underscores the potential for zoonotic transmission between companion animals and humans [11,17,18].

Companion animals remain an overlooked source of zoonotic diseases, with over 70 pathogens capable of infecting humans [1]. National surveys reveal that approximately 46% and 26% of households in the United States own at least one dog or cat, respectively [19], with similar estimates observed in Canadian households [20]. These statistics indicate an inherent risk of zoonotic disease via companion animals, posing an important health concern.

Despite the ability of companion animals to transmit pathogens to humans, they have received limited attention in the context of climate change. At present, our understanding of climate-sensitive zoonotic diseases that are transmissible by companion

animals, the meteorological factors associated with each disease, and the anticipated impacts of changing climates on the burden of these diseases is limited. Given the disparate evidence and lack of current or ongoing reviews concerning this topic, based on a preliminary search of MEDLINE (via Ovid) and Open Science Framework, a systematic synthesis is warranted. These scoping reviews will synthesize existing knowledge regarding how climate change influences the epidemiology of zoonotic diseases that are transmissible by companion animals. The results will identify knowledge gaps and inform future research to monitor, model, and forecast the incidence of zoonotic diseases in companion animal and human populations.

The general aim of these scoping reviews is to map the international evidence of climate-sensitive companion animal zoonotic diseases in cat, dog, and human populations. Because the epidemiology of companion animal zoonotic diseases in these populations differs, separate syntheses are warranted [1]. The current protocol will inform two separate reviews, each focusing on a specific population: one reviewing the available evidence in companion animal populations and another in humans. The objectives are to: 1) identify climate-sensitive companion animal zoonotic diseases, 2) describe associations between meteorological factors and risk of companion animal zoonotic diseases, 3) describe spatiotemporal trends of companion animal zoonotic diseases, and 4) describe the projected impacts of specified future climate emission scenarios on the risk of companion animal zoonotic diseases.

## Materials and methods

### Review questions

What companion animal zoonotic diseases are sensitive to climate change? How is climate change expected to influence the epidemiology of these diseases in companion animal and human populations?

### Population

Epidemiologic studies will be included if the species of interest are cats, dogs, and/or humans.

### Concept

The key concepts are companion animal zoonotic diseases, climate sensitivity, meteorological factors, and climate change, which are defined as follows:

- Companion animal zoonotic diseases: For this scoping review, zoonotic pathogens and diseases associated with companion animals are restricted to those potentially transmitted between cats or dogs and humans, as identified by Weese and Fulford (2011) [1]. For zoonotic diseases with multiple modes of transmission, of which dogs or cats are suspected sources of disease, studies investigating diseases that involve other transmission routes will be included (e.g., salmonellosis).

- Climate sensitivity: A zoonotic pathogen, vector, reservoir, or disease that could be affected by changing climates directly or indirectly and thus are susceptible to changes in their epidemiology [7,21].

- Meteorological factors: *"The average or expected weather for a given location and time."* [22]. *"Weather is the state of atmospheric conditions, including temperature, precipitation, humidity, wind, and extreme events."* [22].

- Climate change: *"A long-term shift in average regional weather conditions, typically occurring over several decades or longer."* [22].

### Context

Primary research articles published in English will be included. The scoping review will maintain an open context without restrictions on geographic location, time, or socio-demographic characteristics.

## Inclusion criteria

- Infections of companion animal zoonotic diseases in cats, dogs, and/or humans are the outcome of interest. Infection can include the presence of disease or positive tests for disease in subclinical individuals.

- Meteorological factors are the primary exposure of interest. Given that climate is defined by long-term meteorological patterns, this inclusion criterion aims to capture short-term relationships between disease and meteorological factors which may provide insight into how a changing climate could impact disease risk.

- Studies that examine the current or projected impacts of meteorological factors on companion animal zoonotic diseases in cats, dogs, and/or humans.

- Primary research articles, including observational (i.e., cross-sectional, case-control, and cohort), descriptive time-series studies, descriptive spatial (or cluster detection) studies, experimental studies, and predictive studies using different climate emission scenarios.

## Exclusion criteria

- Full-text citation is not available.

- Full-text is not available in English.

- Duplicate articles.

- Knowledge synthesis papers, case series, and clinical reports.

## Methods

The proposed scoping reviews will be conducted following the Joanna Briggs Institute (JBI) methodology [23] for scoping reviews and Preferred Reporting Items Systematic Reviews and Meta-Analyses for Scoping Reviews (PRISMA-ScR) [24]. Individual protocols for companion animal and human populations are registered separately on Open Science Framework.

## Search strategy

Three electronic academic databases, MEDLINE (via Ovid), AGRICOLA (via ProQuest), and Web of Science Core Collection, will be searched to locate eligible studies. Citation tracking will be applied to the reference lists of articles included in the scoping review to identify further research. Search results will be imported to Covidence™ for de-duplication, article screening, and data extraction.

A limited initial search of MEDLINE (via Ovid) and Web of Science was conducted to identify articles relevant to climate-sensitive companion animal zoonotic diseases in cat, dog, and human populations. Using the companion animal zoonotic diseases identified by Weese and Fulford (2011) [1] and relevant articles retrieved through this initial search, a comprehensive list of index terms and keywords was developed (S1 File). The proposed search strategy will be applied to each database to retrieve both published academic research and grey literature, adjusting the search as necessary to conform with database syntaxes. An example of the search in MEDLINE (via Ovid) is available in S2 File. The search strategy was further refined through consultations with the broader research team and a research librarian at the University of Guelph. The search strategy will not include publication date or language restrictions.

## Study selection

In Covidence™, independent reviewers will screen the titles and abstracts against the inclusion and exclusion criteria (Level 1), followed by a full-text screening of selected articles (Level 2). Screening questions for title/abstract and full-text

screening are included in this protocol to support agreement between reviewers. Data will then be extracted, analyzed, and presented in graphical and tabular formats.

Disagreements between reviewers at each stage of the selection process will be resolved through discussion. An additional reviewer will be consulted if consensus cannot be reached. Agreement between reviewers will be determined using Cohen's kappa coefficient statistic (κ). A κ statistic of equal to or greater than 0.7 will be considered an acceptable level of agreement between reviewers. The pilot screening will be conducted with 50 articles for Level 1 screening, 20 articles for Level 2 screening, and 5 articles for data extraction. After pilot screening is complete, reviewers will address any conflicts in throughout the screening process by discussion. An additional reviewer will be consulted if consensus cannot be reached.

The following questions will be used for primary, Level 1 screening of article titles and abstracts. If reviewers respond "no" to any of the following questions, articles will be excluded from this review.

1. Is the title/abstract in English?

2. Does the title/abstract describe research that used primary or secondary data?

3. Does the title/abstract consider companion animal zoonotic diseases in cats, dogs, or humans as an outcome of interest?

4. Does the title/abstract describe the impact of a climatic factor on companion animal zoonotic diseases in cats, dogs, or humans?

The following questions will be used for secondary, Level 2 screening of full-text articles. If reviewers respond "no" to any of the following questions, articles will be excluded from this review.

1. Is the full-text provided in English?

2. Does the study use an observational or experimental study design?

3. Does the full-text consider companion animal zoonotic diseases in cats, dogs, or humans as an outcome of interest?

4. Does the full-text describe the impact of current or future climatic factors on the risk of companion animal zoonotic diseases in cat, dog, or human populations?

### Data extraction

After eligibility screening, reviewers will independently extract data from the full-text of relevant articles using Covidence™ and a data extraction form (S3 File). Reviewers will also conduct a pilot test on the *a priori* data extraction form using the first 5 articles to ensure clarity in wording and that the desired data elements for addressing the review questions are captured. The authors of relevant articles will not be contacted to request missing data. As the scoping reviews evolve, the data extraction form will be iteratively updated, if necessary, through regular meetings between reviewers. The reviewers will address any conflicts through discussion. An additional reviewer will be consulted if consensus cannot be reached.

### Data presentation

Results will be presented in two publications: one regarding companion animal health outcomes and another on human health outcomes. In the final scoping reviews, PRISMA-ScR flow diagrams will depict the results of the search strategy and study selection processes [24]. The data extracted from relevant articles will be depicted through a descriptive summary, using a combination of figures, tables, and text, as appropriate. Findings from relevant articles will also be stratified by climate zones (i.e., tropical, subtropics, temperate, and cold). The authors will target Open Access journals to promote accessibility for various organizations and audiences.

## Status and timeline

We are currently undergoing Level 1 screening for both reviews and anticipate that these reviews will be completed by the end of 2025.

## Discussion

Climate change is a complex issue affecting global health through various pathways, including its impact on zoonotic diseases [2,9]. Given the projected changes in global temperatures and other meteorological factors, it is crucial to gain a better understanding of how current and projected meteorological factors affect the epidemiology of zoonotic diseases that can be transmitted by companion animals. The proposed scoping reviews aim to synthesize the available evidence for climate-sensitive zoonotic diseases in dogs, cats, and human populations worldwide, particularly 1) spatiotemporal patterns, 2) meteorological risk factors, and 3) future disease trends due to climate change. The findings will enrich the evidence on companion animal zoonotic diseases worldwide, inform future research directions, and highlight considerations for health system planning in the context of climate change.

## Limitations

There are several limitations that should be acknowledged. First, only articles published in the English language will be included. Second, despite consulting Weese and Fulford (2011) [1] to identify a list of companion animal zoonotic diseases, the search strategy did not include keywords for every zoonotic pathogen that can be transmitted between dogs, cats, and humans. Third, despite a systematic search of academic literature, relevant sources may be unintentionally omitted such that not all available literature on climate-sensitive companion animal zoonotic diseases will be captured.

## Supporting information

**S1 File. Appendix 1: Proposed keywords and controlled vocabulary for the key concepts of meteorological factors, companion animals, human health, and companion animal zoonotic diseases.**
(DOCX)

**S2 File. Appendix II Search terms for MEDLINE via OVID, executed February 1, 2024.**
(DOCX)

**S3 File. Appendix III: Proposed data extraction form.**
(DOCX)

**S1 Data. PRISMA-P-checklist - PloS One – Protocol.**
(DOCX)

## Acknowledgments

Thank you to Jacqueline Kreller-Vanderkooy at the University of Guelph for her contributions to the development of the search strategy for this protocol.

## Author contributions

**Conceptualization:** Lauren E. Grant.

**Funding acquisition:** Lauren E. Grant.

**Methodology:** Samantha Hobson, John Mallare, Heather Grieve, J. Scott Weese, Lauren E. Grant.

**Supervision:** Lauren E. Grant.

**Writing – original draft:** Samantha Hobson, John Mallare.

**Writing – review & editing:** Heather Grieve, J. Scott Weese, Lauren E. Grant.

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
