## [Decision Letter · Decision Letter 0]

Dear Dr. Grant,

Thank you for submitting your manuscript to PLOS ONE. After careful consideration, we feel that it has merit but does not fully meet PLOS ONE’s publication criteria as it currently stands. Therefore, we invite you to submit a revised version of the manuscript that addresses the points raised during the review process.

Reviewer 1: 1. This article is a protocol for an in-process scoping review. The review topic is interesting – the impact of climate change on zoonotic diseases in companion animal (dog and cat) and human populations.

2. The authors describe using the protocol to satisfy two separate reviews – one for evidence in companion animal populations, and one for human populations. There is probably a good reason for doing this, but they do not explain it in the protocol. It would be helpful if this was added.

3. I feel like the stated inclusion criteria on page 6 needs to weave in climate change into the meteorological factor(s) as the primary exposure. I know it is implied, but without explicit statement, it appears as though studies that simply focus on weather could be included, and I am not sure if that is the intent. I suggest this be justified/explained or amended.

4. Exclusion criteria – from this, it appears that the review will include theses, dissertations, conference proceedings and abstracts. Is this the intent? In addition, the search strategy states that grey literature will be included, but there are no specific inclusion/exclusion criteria stated for what grey literature will be included. For example, it appears that media would be excluded (which is good). This needs to be clarified. It is mentioned under strategy, but they should be part of exclusion criteria.

5. First level screening – “unsure” is implied to move onto level 2, because it is not “no”. What happens in level 2 screening if the answer is “unsure”?

6. Second level screening – why does the second level screening not include question 3 from level one?

7. I see the list of index search terms and key words in Appendix I. I do not see Appendix II with the database syntaxes. It would be nice to see the database-specific search strategies (+/- results).

Reviewer 2: It would be helpful to have the searches included as an appendix here. It isn't possible to evaluate the search for completeness as it currently stands. However, given that the paper is well described otherwise, I don't think this is an issue for the protocol. When publishing the final review, be sure to follow PRISMA-S reporting guidelines so the search is transparent and reproducible.

Academic Editor’s Comments: The experiences with influenza viruses’ seasonality indicate significant differences in terms of the time of onset, duration, number of peaks, and amplitude of epidemics between temperate and tropical/subtropical regions. The patterns are highly diverse in tropical countries and may be out of phase with the WHO recommendations for their respective hemisphere.  Therefore, suggest including hemispheric variations between temperate and tropical/subtropical regions as one of the inclusion criteria as an evidence for health systems planning.

We look forward to receiving your revised manuscript.

Kind regards,

Asokan Govindaraj Vaithinathan

Academic Editor

PLOS ONE

Journal Requirements:

Additional Editor Comments:

Thank you for choosing an important and timely topic for a scoping review.

We have received the reviewers' comments and is given below for response and suggest revising the manuscript accordingly and resubmission.

Reviewer 1: 1. This article is a protocol for an in-process scoping review. The review topic is interesting – the impact of climate change on zoonotic diseases in companion animal (dog and cat) and human populations.

2. The authors describe using the protocol to satisfy two separate reviews – one for evidence in companion animal populations, and one for human populations. There is probably a good reason for doing this, but they do not explain it in the protocol. It would be helpful if this was added.

3. I feel like the stated inclusion criteria on page 6 needs to weave in climate change into the meteorological factor(s) as the primary exposure. I know it is implied, but without explicit statement, it appears as though studies that simply focus on weather could be included, and I am not sure if that is the intent. I suggest this be justified/explained or amended.

4. Exclusion criteria – from this, it appears that the review will include theses, dissertations, conference proceedings and abstracts. Is this the intent? In addition, the search strategy states that grey literature will be included, but there are no specific inclusion/exclusion criteria stated for what grey literature will be included. For example, it appears that media would be excluded (which is good). This needs to be clarified. It is mentioned under strategy, but they should be part of exclusion criteria.

5. First level screening – “unsure” is implied to move onto level 2, because it is not “no”. What happens in level 2 screening if the answer is “unsure”?

6. Second level screening – why does the second level screening not include question 3 from level one?

7. I see the list of index search terms and key words in Appendix I. I do not see Appendix II with the database syntaxes. It would be nice to see the database-specific search strategies (+/- results).

Reviewer 2: It would be helpful to have the searches included as an appendix here. It isn't possible to evaluate the search for completeness as it currently stands. However, given that the paper is well described otherwise, I don't think this is an issue for the protocol. When publishing the final review, be sure to follow PRISMA-S reporting guidelines so the search is transparent and reproducible.

Academic Editor’s Comments: The experiences with influenza viruses’ seasonality indicate significant differences in terms of the time of onset, duration, number of peaks, and amplitude of epidemics between temperate and tropical/subtropical regions. The patterns are highly diverse in tropical countries and may be out of phase with the WHO recommendations for their respective hemisphere. Therefore, suggest including hemispheric variations between temperate and tropical/subtropical regions as one of the inclusion criteria as evidence for health systems planning.

Reviewers' comments:

Reviewer's Responses to Questions

**Comments to the Author**

1. Does the manuscript provide a valid rationale for the proposed study, with clearly identified and justified research questions?

Reviewer #1: Yes

Reviewer #2: Yes

2. Is the protocol technically sound and planned in a manner that will lead to a meaningful outcome and allow testing the stated hypotheses?

Reviewer #1: Yes

Reviewer #2: Yes

3. Is the methodology feasible and described in sufficient detail to allow the work to be replicable?

Reviewer #1: Yes

Reviewer #2: No

4. Have the authors described where all data underlying the findings will be made available when the study is complete?

Reviewer #1: Yes

Reviewer #2: Yes

5. Is the manuscript presented in an intelligible fashion and written in standard English?

Reviewer #1: Yes

Reviewer #2: Yes

You may also provide optional suggestions and comments to authors that they might find helpful in planning their study.

Reviewer #1: 1. This article is a protocol for an in-process scoping review. The review topic is interesting – the impact of climate change on zoonotic diseases in companion animal (dog and cat) and human populations.

2. The authors describe using the protocol to satisfy two separate reviews – one for evidence in companion animal populations, and one for human populations. There is probably a good reason for doing this, but they do not explain it in the protocol. It would be helpful if this was added.

3. I feel like the stated inclusion criteria on page 6 needs to weave in climate change into the meteorological factor(s) as the primary exposure. I know it is implied, but without explicit statement, it appears as though studies that simply focus on weather could be included, and I am not sure if that is the intent. I suggest this be justified/explained or amended.

4. Exclusion criteria – from this, it appears that the review will include theses, dissertations, conference proceedings and abstracts. Is this the intent? In addition, the search strategy states that grey literature will be included, but there are no specific inclusion/exclusion criteria stated for what grey literature will be included. For example, it appears that media would be excluded (which is good). This needs to be clarified. It is mentioned under strategy, but they should be part of exclusion criteria.

5. First level screening – “unsure” is implied to move onto level 2, because it is not “no”. What happens in level 2 screening if the answer is “unsure”?

6. Second level screening – why does the second level screening not include question 3 from level one?

7. I see the list of index search terms and key words in Appendix I. I do not see Appendix II with the database syntaxes. It would be nice to see the database-specific search strategies (+/- results).

Reviewer #2: It would be helpful to have the searches included as an appendix here. It isn't possible to evaluate the search for completeness as it currently stands. However, given that the paper is well described otherwise, I don't think this is an issue for the protocol. When publishing the final review, be sure to follow PRISMA-S reporting guidelines so the search is transparent and reproducible.

**Do you want your identity to be public for this peer review?** For information about this choice, including consent withdrawal, please see our Privacy Policy

Reviewer #1: **Yes: ** Simon Otto

Reviewer #2: No

---

## [Author Response · Author response to Decision Letter 1]

8 May 2025

May 6, 2025

Thank you for the opportunity to submit a revised version of our manuscript in response to the editor and reviewer comments. We are grateful for their time and effort to provide us with feedback that has undoubtedly enhanced the quality of this manuscript. Please find our point-by-point response below:

Reviewer 1:

1. This article is a protocol for an in-process scoping review. The review topic is interesting – the impact of climate change on zoonotic diseases in companion animal (dog and cat) and human populations. Thank you for your feedback.

2. The authors describe using the protocol to satisfy two separate reviews – one for evidence in companion animal populations, and one for human populations. There is probably a good reason for doing this, but they do not explain it in the protocol. It would be helpful if this was added. This has been justified in the introduction, lines 83-87.

3. I feel like the stated inclusion criteria on page 6 needs to weave in climate change into the meteorological factor(s) as the primary exposure. I know it is implied, but without explicit statement, it appears as though studies that simply focus on weather could be included, and I am not sure if that is the intent. I suggest this be justified/explained or amended. This has been clarified on lines 124-127.

4. Exclusion criteria – from this, it appears that the review will include theses, dissertations, conference proceedings and abstracts. Is this the intent? Thank you for your feedback. The original intent was to include theses and dissertations. This change has been reflected on line 138.

In addition, the search strategy states that grey literature will be included, but there are no specific inclusion/exclusion criteria stated for what grey literature will be included. For example, it appears that media would be excluded (which is good). This needs to be clarified. It is mentioned under strategy, but they should be part of exclusion criteria. Inclusion and exclusion criteria have been updated to reflect this feedback. We ultimately elected to focus on peer-reviewed evidence only, excluding grey literature from our search.

5. First level screening – “unsure” is implied to move onto level 2, because it is not “no”. What happens in level 2 screening if the answer is “unsure”? Thank you for your question. At level 2 screening, the reviewer will read the full article and will be able to confirm if the article meets the inclusion criteria. Two reviewers will complete each level of screening. Disagreements/ uncertainty regarding the inclusion criteria will be discussed between reviewers, especially when conflicts rise. A third reviewer would be consulted in the event the conflict could not be resolved. This has been clarified on lines 172-173.

6. Second level screening – why does the second level screening not include question 3 from level one? Question 3 has been added to level 2 screening for clarity (lines 188-189)

7. I see the list of index search terms and key words in Appendix I. I do not see Appendix II with the database syntaxes. It would be nice to see the database-specific search strategies (+/- results). Appendix II has been updated to include a MEDLINE (via Ovid)-specific search strategy. The final review will include +/- results, as indicated by PRISMA-S guidelines.

----

Reviewer 2: It would be helpful to have the searches included as an appendix here. It isn't possible to evaluate the search for completeness as it currently stands. However, given that the paper is well described otherwise, I don't think this is an issue for the protocol. When publishing the final review, be sure to follow PRISMA-S reporting guidelines so the search is transparent and reproducible. Thank you for your feedback. Appendix II has been updated to include a MEDLINE (via Ovid)-specific search. This was also clarified on lines 156-157, and line 336. We intend to include separate searches for each database in our final publications, as suggested by PRISMA-S guidelines.

Academic Editor’s Comments: The experiences with influenza viruses’ seasonality indicate significant differences in terms of the time of onset, duration, number of peaks, and amplitude of epidemics between temperate and tropical/subtropical regions. The patterns are highly diverse in tropical countries and may be out of phase with the WHO recommendations for their respective hemisphere. Therefore, suggest including hemispheric variations between temperate and tropical/subtropical regions as one of the inclusion criteria as an evidence for health systems planning. Thank you for your feedback. This will be done for the review focused on human populations and is reflected in the individual protocol referenced on line 43.

We look forward to a favourable response.

---

## [Editor Report · Decision Letter 1]

Climate-sensitive zoonotic diseases transmissible by companion animals: A scoping review protocol

PONE-D-24-20742R1

Dear Dr.Grant,

We’re pleased to inform you that your manuscript has been judged scientifically suitable for publication and will be formally accepted for publication once it meets all outstanding technical requirements.

Kind regards,

Asokan Govindaraj Vaithinathan

Academic Editor

PLOS ONE
---

## [Editor Report · Acceptance letter]

PONE-D-24-20742R1

PLOS ONE

Dear Dr. Grant,

I'm pleased to inform you that your manuscript has been deemed suitable for publication in PLOS ONE. Congratulations! Your manuscript is now being handed over to our production team.

Kind regards,

on behalf of

Dr. Asokan Govindaraj Vaithinathan

Academic Editor

PLOS ONE